# Trends and Causes of Regional Income Inequality in China

**Xiao Yan and Saidatulakmal Mohd ***

School of Social Sciences, Universiti Sains Malaysia, Gelugor 11800, Penang, Malaysia
* Correspondence: eieydda@usm.my

**Abstract:** Although China has been successful in reducing national income inequality over the past decade, regional income inequality shows a fluctuating trend. The pre-tax income shares of the top 10% and 1% have grown since 1978 and reached about 40% and 15% of the total income share in 2015. Meanwhile, the pre-tax income shares of the bottom 50% have been falling, having dropped from one-quarter of the total income share to less than that of the top 1% in 2015. With this background, this study investigated the trends of income inequality from 2000 to 2020 in west, central, northeast, and east regions in China and analysed their influence factors. Income data from 271 prefecture-level cities in mainland China between 2010 and 2019 were used to calculate the Theil index in each province and region. The analysis was segregated based on urban and rural areas in four regions: east, northeast, central, and west. The Theil index indicated that the income inequality of different regions in China showed a declining trend in rural areas, and a declining and then increasing trend in urban areas. Furthermore, local economic development has a positive impact on income inequality, whereas the urbanization rate and fiscal spending rate have negative impacts on income inequality.

**Keywords:** income inequality; spatial disparity; Theil index





## 1. Introduction

The concept of income inequality was proposed by Ricardo (1817) in the theory of factor income distribution [1]. Pareto (1897) defined income inequality by measuring the proportion of income in different economic units using the Pareto accumulative distribution function at the micro level [2]. Income inequality refers to the gap caused by the difference between high- and low-income levels or the difference in the proportion of income occupied, which corresponds to income equality.

According to Henderson [3], inequality persists in all economies, even in communist ones such as the Soviet Union. He also draws attention to how political influence and resource distribution create patterns of inequality within a society. Regional income inequality is one of the major characteristics of both developed and developing countries. Empirical studies supported by data and theory posit that the income inequality trends in most Western countries follow a U-shape in the long term [4,5]. Thanks to globalization and technological development, the growth of inequality in the world has recently shown a slowing trend, which is closely related to the development of emerging countries and the reduction in international inequality [6–8]. Most research on income inequality in emerging countries, such as China and India, is devoted to examining the Kuznets inverted U-shaped trend, which indicates that income disparity among individuals tends to decrease over time as the economy grows [9–12]. Inequality undermines sustainability, which could lead to a variety of negative outcomes such as resource depletion, social unrest, and environmental degradation that could affect future generations [13]. Unsustainable policies could lead to wealth creation for a handful of the population and further exacerbate inequality. Addressing income inequalities within and between countries is seen as a prerequisite for the attainment of the Sustainable Development Goals (SDGs) [14].

China, as an emerging country with the largest population in the world, has accomplished a key breakthrough in reducing international income inequality. China's economic

development has experienced rapid and stable growth in the past two decades, one of the factors in reducing international inequality [9,10]. As shown in Figure 1, the GDP of different regions, an indicator reflecting regional economic growth, increased from 2000 to 2020. More specifically, economic growth is most obvious in the east, where the GDP has risen tenfold in 20 years. The central and western regions have similar economic growth rates, while the central region has a relatively fast speed. The GDP of the west increased to 222.43 billion yuan in 2020. The economic growth of northeast China is the smallest, increasing from 1.13 billion yuan in 2000 to 5.31 billion yuan in 2020; simultaneously, residents' income experienced a similar trend. The income of the eastern region occupies a dominant position in the national income, reaching five times that of the northeastern region in 2020. It is worth noting that although residents' income in the central and western regions has long been at a similar level, the western region surpassed the central region for the first time in 2020.

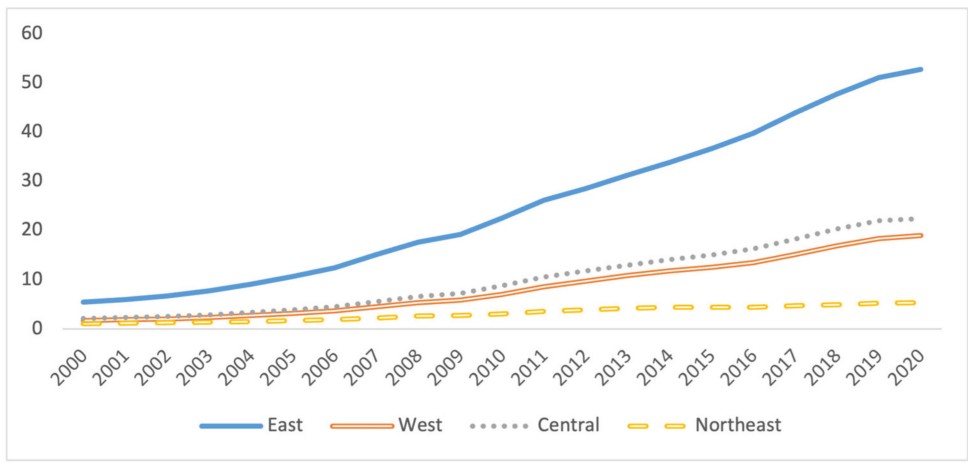

**Figure 1.** Income from 31 provinces, 2000–2020. Source: National Bureau of Statistics of China, 2022.

As economic growth in China has accelerated since 1978, rising intranational income inequality has attracted scholars' attention [15–17]. According to data from the World Inequality Database, the pre-tax income shares of the top 10% and 1% have grown since 1978 and reached about 40% and 15% of the total income share in 2015. At the same time, the pre-tax income shares of the bottom 50% have been falling, dropping from one-quarter of the total income share to less than that of the top 1% in 2015. The importance of China's income inequality in determining the global inequality trend is widely discussed [10,16]. Although the official data indicate a slight fall in national income inequality over the past ten years, whether regional income inequality has declined in recent years is worth investigating. Additionally, it is valuable to explore the drivers of changes in income inequality trends.

According to previous research, there are mainly three stages of personal income convergence across different regions. The first stage was in the late 1970s and 1980s, when China began its reform and opening up, during which regional income increased dramatically with economic growth [16,18–20]. The second stage began in 1991, as regional disparity increased steadily, reaching its peak around 2010 [3,15]. The last stage is after 2010, and empirical evidence from macro and micro databases indicates regional income convergence [21,22].

China's spatial income inequality can be defined by inequality among regions and urban–rural income disparity [23–25]. Certain regions, especially in eastern China, have more disproportionate advantages from the reform and opening up because of preferential policies, natural endowment, and improved infrastructure [15]. Compared with the central and western areas, the income level in the east is higher, resulting in income inequality among regions. Inequality among regions and urban-rural income disparity are not entirely

independent, suggesting that the difference in regional development also promotes the further differentiation of urban and rural development levels [3].

Therefore, this paper investigates the trends of income inequality from 2000 to 2020 in the west, central, northeast, and east regions of China and analyses the influential factors linked to the different development characteristics of the regions.

Due to the imbalance between urban and rural development, the manifestation and drivers of personal income inequality in both areas could also be varied. In this paper, we divide the objective population into two parts: people who live in urban areas and rural areas. Based on the above background, this paper has two main objectives: (i) to investigate the nature of regional income inequality in China; and (ii) to understand the factors influencing income inequality in China.

## 2. Materials and Methods

### 2.1. One-Stage Theil Index Decomposition

Compared to the more traditional Gini coefficient, the Theil index is more sensitive to income transfers from the poor to the rich and therefore outperforms the Gini coefficient in terms of intra-group decomposition [26,27]. Theil [28] expanded the information theory by Shannon (1948) to measure the share of residents' income in a country [28]. It provides a tool to measure equality. Shannon's information content reaches the maximum level when the income tends to be equal. The standard one-stage Theil inequality decomposition method is introduced as follows.

If we have $y_i$ to represent individual income, and the population size is $n$, then we can obtain the average income $\mu_Y = \frac{\sum_{i=1}^{n} y_i}{n}$. Assuming that the ratio of the province $j$ population is $\theta_n$, we can derive $\theta_n = \frac{n_{ij}}{n}$. The Theil index ($L$) can also measure the overall regional income inequality, employing population shares as weights:

$$L_p = \sum_{i,j} \theta_n . \log \frac{\theta_n}{\mu_y} \tag{1}$$

If we have $y$ as the total income of a country and $y_{ij}$ as the income of province $j$ in region $i$, we obtain $y = \sum_{i,j} y_{ij}$. Then, we can assume that $n$ is the total population of the country and $n_{ij}$. The relationship between these two indicators is $n = \sum_{i,j} n_{ij}$.

If we have $L_{pi}$ to measure income inequality among different provinces (between-province inequality) in region $i$, the $L_{pi}$ can be calculated as follows:

$$L_{pi} = \sum_{j} \theta_{ni} . \log \frac{\theta_{ni}}{\mu_{yi}} \tag{2}$$

where $\mu_{yi} = \frac{y_{ij}}{y_i}$ and $\theta_{ni} = \frac{n_{ij}}{n_i}$.

We apply this between-province inequality $L_{pi}$ to the overall inequality. Then, we can obtain the one-stage income exponential decomposition of this province:

$$L_p = \sum_{i} \theta_n . L_{pi} + \sum_{i} \theta_n . \log \frac{\theta_n}{\mu_y} \tag{3}$$

$$L_{BR} = \sum_{i} \theta_n . \log \frac{\theta_n}{\mu_y} \tag{4}$$

$$L_{WR} = \sum_{i} \theta_n . L_{pi} \tag{5}$$

where $L_{BR}$ is the measurement of income inequality between regions, and $L_{WR}$ is the measurement of income inequality within regions. This component is a weighted average of

the between-province income inequality ($L_{pi}$) for each region. It can be concluded that overall income inequality consists of within-region inequality and between-region inequality:

$$L_p = L_{WR} + L_{BR} \tag{6}$$

At the same time, the Theil index can also measure the interregional contribution rate and the intraregional contribution rate:

$$W_{ir} = (Y_i/Y) * (L_{WR}/L) \tag{7}$$

$W_{ir}$ repents the contribution rate of region $i$ to the overall Theil index, reflecting the impact of regional differences on the overall disparities.

### 2.2. Empirical Model on the Factors Influencing Income Inequality

Kuznets (1955) described changes in income disparities over time in different levels of economic situations among countries [29]. The hypothesis states that during the process of economic growth, the disparities between rich and poor first widen and then narrow. This pattern is known as the Kuznets curve, and it suggests that the dividends of economic growth would spread to the bottom as industrialization develops [30]. Kuznets (1955) attributed the narrowing of the income distribution disparity to three main reasons [29]. First, the development of education amid economic growth leads to an improvement in the quality of the workforce and an increase in productivity [31,32]. This leads to an increase in the relative price of labour, resulting in a rise in the share of labour income in national income and a fall in the share of property income [33,34]. Secondly, with the upgrading of the industrial structure, the gap in labour output between the agricultural, industrial, and service sectors narrows [25]. In particular, the decline in the share of agricultural labour narrows the gap in average labour output between sectors, resulting in a narrower income distribution gap [35–37]. Thirdly, the structure of the workforce has changed, such as the upward trend in the share of blue-collar workers in the total workforce and the decline in the share of white-collar workers, which has led to a reduction in the number of lower-income earners and a lessening of income inequality [29,38,39].

The panel data model can reflect the variation rules and characteristics of variables in both cross-section and time two-dimensional space. Therefore, it can reflect the income inequality in different provinces in various periods. Based on the Kuznets Theory, the income inequality in a country is associated with economic growth, human capital development, and financial condition improvement. The spatial income inequality should be considered to impact income inequality. In this paper, we use the urbanization rate, GDP growth rate, education growth rate, higher education rate, fiscal spending/GDP, and FDI/GDP as the dependent variables. The panel data model constructed in this paper is as follows:

$$Y = \beta_0 + \beta x_{it} + \varepsilon_{it} \tag{8}$$

$Y$ is the Theil index for each province; $x_{it}$ represents the dependent variables; $\beta$ is a vector of coefficients on $x_{it}$; $\beta_0$ is the constant term; $\varepsilon_{it}$ is a random error term.

When dealing with panel data, we were faced with the choice of a fixed effects model, a random effects model, and an OLS model. The Hausman test was carried out and the test results were obtained by Stata 17 software. The results are discussed in the next section.

### 2.3. Data

To achieve the first objective of this research, we used two types of data: (i) macro income data from 31 provinces for the years between 2000 and 2020 to estimate the trend of income inequality based on provinces and regions; and (ii) micro income data from 271 cities in 31 provinces in mainland China between 2010 and 2019 to estimate the trend of income inequality between urban and rural areas. These two datasets were used because

limited income data are available to analyse urban–rural income disparity in China. To achieve our second objective, we used province-level data between 2000 and 2020.

The analysis was segregated based on urban and rural areas in four regions: east, northeast, central, and west (Table 1). The Chow test results showed that there were indeed significant differences in the regression model parameters between different regions. The Chow test statistic was 25.37 and the *p*-value was less than 0.05, indicating that we can reject the null hypothesis that the model parameters are the same across different regions.

**Table 1.** Eastern, northeastern, central, and western regions' division.

| Region | Province |
|---|---|
| East | Beijing, Tianjin, Hebei, Shanghai, Jiangsu, Zhejiang, Fujian, Shandong, Guangdong, Hainan |
| Northeast | Liaoning, Jilin, Heilongjiang |
| Central | Shanxi, Anhui, Jiangxi, Henan, Hubei, Hunan |
| West | Neimenggu, Guangxi, Sichuan, Guizhou, Yunnan, Chongqing, Shanxi, Ganxu, Qinghai, Ningxia, Xinjiang, Tibet |

Source: China's National Bureau of Statistics (NBS), [40]

The variables used for the analysis are shown in Table 2 and their descriptive statistics are shown in Table 3.

**Table 2.** Variables and their descriptions.

| Notation | Variable Description | Expected Signs | Source |
|---|---|---|---|
| GDP | GDP growth rate | $+/-$ | Kuznets Theory |
| EDU | Growth rate of average years of schooling | $+/-$ | [15] |
| HEDU | Tertiary education participation rate | $+/-$ | [41] |
| FISCAL | The ratio of fiscal spending to GDP | $-$ | [42] |
| URBAN | Urbanization rate | $-$ | [43,44] |
| FDI | The ratio of foreign direct investment to GDP | $+/-$ | [45] |

**Table 3.** Descriptive statistics of variables.

| Notation | Obs. | Mean | SD | Min | Max |
|---|---|---|---|---|---|
| GDP | 651 | 0.5093597 | 0.1578866 | 0.193 | 0.896 |
| EDU | 651 | 0.1264653 | 0.0631229 | $-0.0810241$ | 0.5074087 |
| HEDU | 651 | 0.000352 | 0.0408615 | $-0.2089615$ | 0.1761812 |
| FISCAL | 651 | 0.1017882 | 0.0717434 | 0.0078969 | 0.5048593 |
| URBAN | 651 | 0.2431599 | 0.1827797 | 0.0691256 | 1.353777 |
| FDI | 651 | 0.5093597 | 0.1578866 | 0.193 | 0.896 |

Source: Authors own calculation using data sources from China's National Bureau of Statistics (NBS), [40].

The population situation for each region is shown in Figure 2. Population in the east is dominant and the growth rate is the most rapid, reaching 0.59 billion in 2020. The western and central regions are roughly equal in population, with the northeast having the smallest population.

As for the individual income of urban and rural residents, Figure 3 shows urban residents' income per capita for east, west, central, and northeast regions, and Figure 4 is the income per capita for rural residents in the east, west, central, and northeast regions.

Personal disposable income levels in all regions showed a gradual upward trend from 2010 to 2019. The per capita income of urban residents is higher than that of rural residents in all regions. Eastern urban areas have the highest per capita income between 2000 and 2020, more than doubling from 21,716.6 in 2010. Meanwhile, rural areas in western regions have the least per capita income, fluctuating around almost one-quarter of that in the eastern urban areas. With the gradual increase in the urbanization rate, more people from rural areas are moving to urban areas, and the total disposable income in each region is closer to the income of urban residents. Therefore, studies of total per capita income inequality do not reflect changes in urban and rural incomes. This paper uses the per capita

income in 271 cities of 4 regions to investigate the regional inequality changes in urban and rural areas.

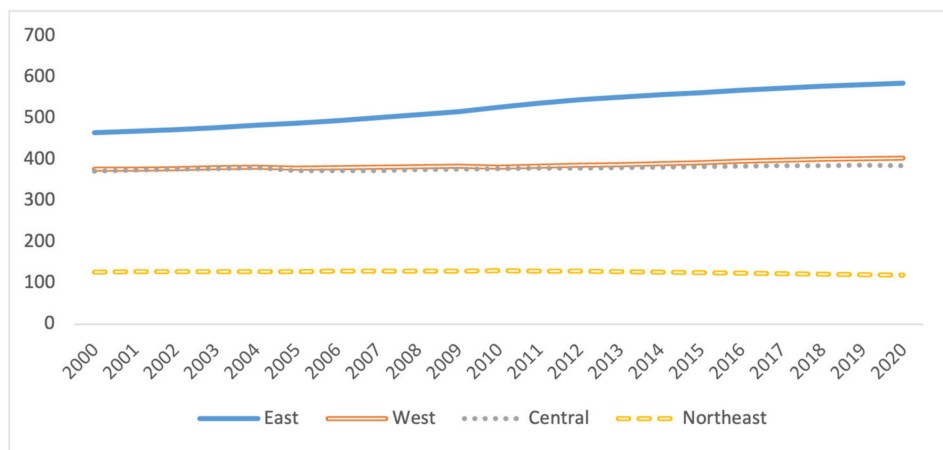

**Figure 2.** Population in different regions in China, 2000–2020. Source: China's National Bureau of Statistics (NBS), [40].

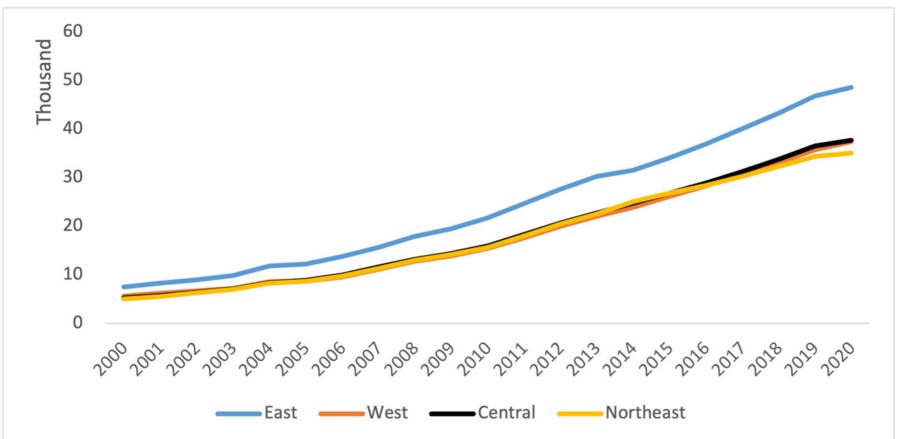

**Figure 3.** Urban income per capita in east, west, central, and northeast regions. Source: China's National Bureau of Statistics, (NBS), [40].

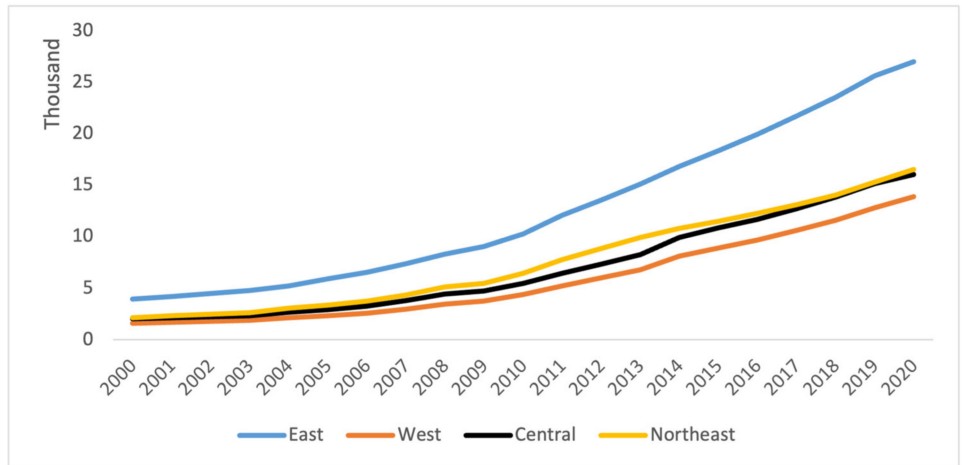

**Figure 4.** Rural income per capita in east, west, central, and northeast regions. Source: China's National Bureau of Statistics, (NBS), [40].

There are additional macro-economic data from the National Bureau of Statistics of China between 2000 and 2020, including urbanization rate, GDP growth rate, education growth rate, higher education rate, fiscal spending, and foreign direct investment in 31 provinces of 4 regions. These data can be used to estimate the influencing factors of regional income inequality.

## 3. Results

### 3.1. Trends of Income Inequality According to Provinces and Regions

As shown in Figure 5, the change in income inequality in China's provincial administrative regions, as measured by the Theil index, shows a trend of rising first and then decreasing from 2000 to 2020. Overall, the change in income inequality in the western region is the largest, while that in the eastern and northeastern regions is relatively small.

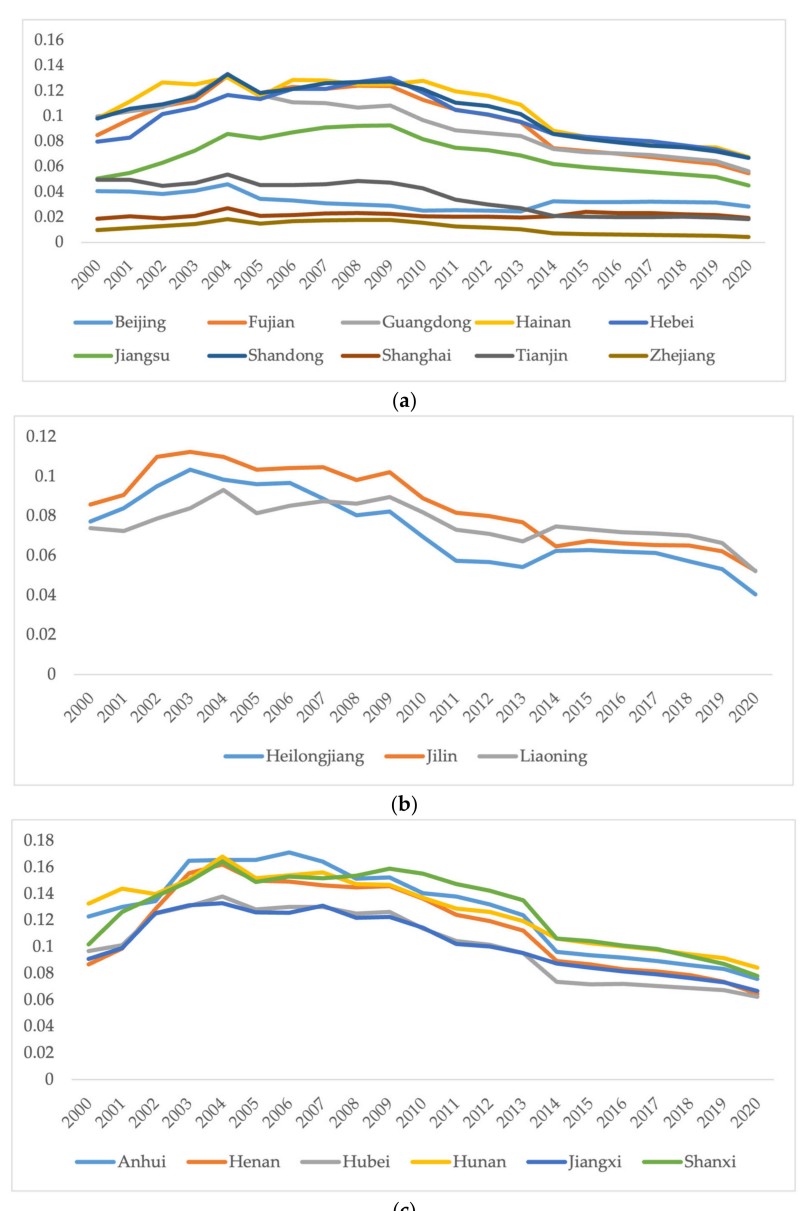

(a)

(b)

(c)

**Figure 5.** *Cont.*

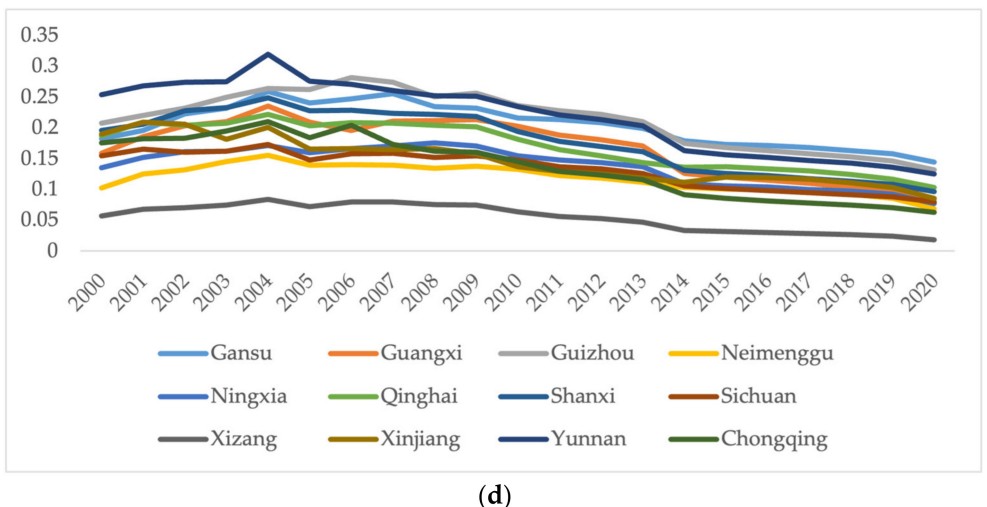

(**d**)

**Figure 5.** Provincial Theil index in east, northeast, central, and west China: (**a**) Theil index for provinces in eastern region; (**b**) Theil index for provinces in northeast region; (**c**) Theil index for provinces in central region; (**d**) Theil index for provinces in western region. Source: Authors' own calculation using data source from China's National Bureau of Statistics, (NBS), [40].

Since 2000, the income inequality of each provincial administrative region has maintained an eight- to ten-year growth period. The turning point in income inequality was between 2008 and 2010, and it first appeared in the western region, starting with Yunnan, Guizhou, Gansu, and other western provinces. This is associated with the western development strategy implemented by the state in the early 20th century and more financial support for the development of the west.

The trend of declining income inequality spread across the country from 2010 to 2013. Income inequality among provinces in the central region began earlier than in the east and northeast and had higher similarity in the changing trend. In Shanxi Province, for example, the income inequality index decreased from 0.159 by more than a third between 2009 and 2013. Provinces in the northeast also experienced a more pronounced decline, with the Theil index for all provinces in this area falling below 0.008 in 2013. In contrast, provincial administration areas in the eastern region showed more differentiation during this period. On the one hand, some provinces, such as Hainan, Hebei, Jiangsu, Guangdong, and Fujian, showed a downward trend in the Theil index. On the other hand, the Theil index in some municipalities, such as Shanghai and Beijing, remained relatively stable. This may be related to the high level of economic development, urbanization, and other endowment factors in Beijing and Shanghai.

The Theil index of the vast majority of provinces showed a more significant degree of decline after 2013. Income inequality in these provinces, autonomous regions, and municipalities was at its lowest level in 20 years in 2020. For the central and western regions, this relates to the migration of manufacturing from coastal areas, which has boosted incomes and wages. For provinces in eastern China, with the liberalization of the household registration system, more labour has moved from rural to urban areas, which is related to the narrowing of the income gap between rural and urban areas. It has further decreased the total income inequality in different provinces. In contrast, the income disparity in northeast China's Heilongjiang and Liaoning provinces increased from 2013 to 2014, and after 2014, the decrease was relatively small compared with other provinces. The serious population loss and lack of foreign investment make the economic development of northeast China stagnate slightly, affecting the region's income inequality to a certain extent.

### 3.2. Trends of Income Inequality in Urban and Rural Areas

To analyse the changes in income differences between urban and rural areas in various regions, we investigated the urban–rural income gap in 271 cities of 4 regions in China. The income disparity of prefecture-level cities in a region is investigated as the intraregional difference, and the population is used as the weight to calculate the total Theil index in urban and rural areas. The results and trend changes are presented in Figures 6 and 7.

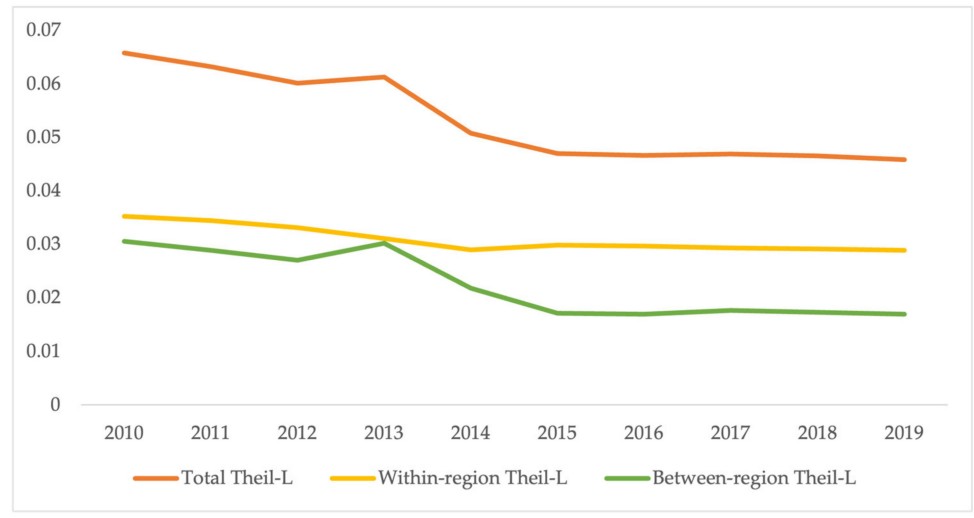

**Figure 6.** Total inequality and decomposition of urban income inequality, 2010–2019. Source: Authors own calculation using data source from China's National Bureau of Statistics, (NBS), [40].

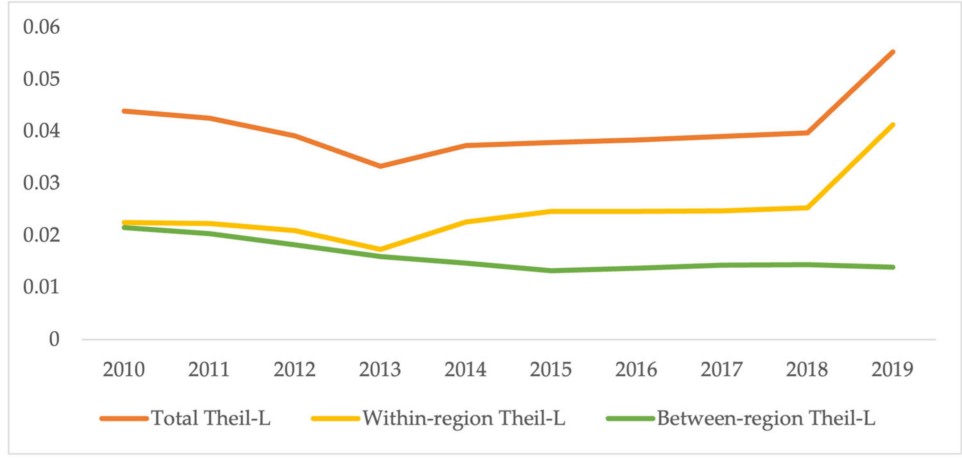

**Figure 7.** Total inequality and decomposition of rural income inequality, 2010–2019. Source: Authors own calculation using data source from China's National Bureau of Statistics, (NBS), [40].

As shown in Figures 6 and 7, the total Theil index of rural areas showed a downward trend, while that of urban areas first showed a downward trend and then an upward trend. Before 2012, the overall inequality between urban and rural areas showed a downward trend. The total Theil index in urban areas decreased from 0.044 in 2010 to 0.039 in 2012, and the total Theil index in rural areas decreased from 0.066 in 2010 to 0.060. After 2013, the rural Theil index experienced a process of first increasing to 0.061 and then decreasing to 0.046, while the urban income gap experienced the opposite. The urban total Theil index rose to 0.055 in 2019, surpassing the rural aggregate Theil index for the first time.

For rural areas, the interregional Theil index curve is consistent with the total Theil index curve, both showing a brief increase from 2012 to 2013. This indicates that although

the differences among the four regions fluctuated slightly, they showed a downward trend on the whole, and were always lower than the intraregional Theil index. In terms of the contribution rate to the overall trend, the contribution rate within the region was almost equal during 2010–2013, and it became greater than that among regions after 2014, indicating that the main reason for the income inequality is the income gaps within the four regions.

For the urban area, the intraregional Theil index curve is consistent with the total Theil index curve, which first shows a decline and then a rise. It was in a declining stage before 2013 and then rose continuously. This shows that the income differences in the four regions gradually narrowed and then expanded. The regional Theil index curve maintains a downward trend and is always lower than the intraregional index. In terms of the contribution rate to the total index, the within-region contribution rate is also greater than that between regions after 2014, indicating that the main reason for the income disparities is the income inequality within the four regions.

### 3.3. Intraregional Disparity

To further examine income inequality in urban and rural areas, we divided the urban and rural areas in the four regions into eastern, western, central, and northeastern urban areas, and eastern, western, central, and northeastern rural areas. We can then calculate the measurement of income inequality between and within regions in both urban and rural areas. The results of the examination are shown in Figures 8 and 9.

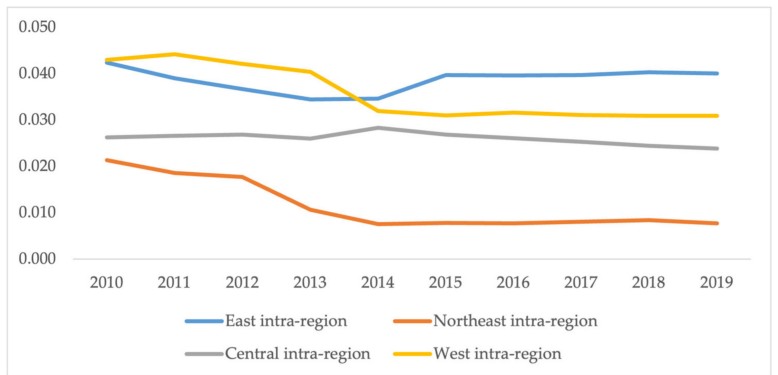

**Figure 8.** Intraregional income inequality in rural areas in the four regions. Source: Authors own calculation using data source from China's National Bureau of Statistics, (NBS), [40].

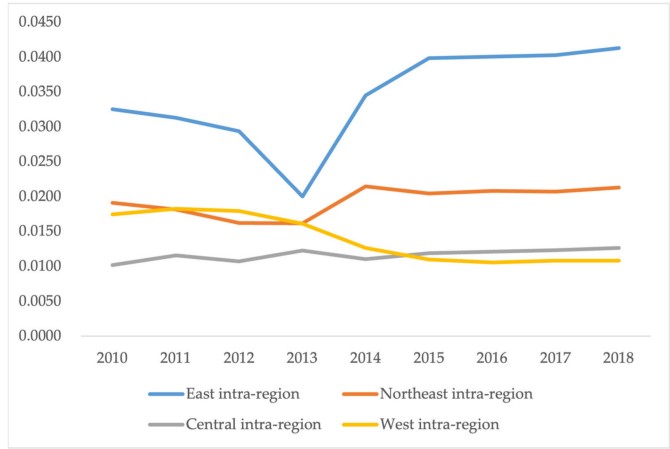

**Figure 9.** Intraregional income inequality in urban areas in the four regions. Source: Authors own calculation using data source from China's National Bureau of Statistics, (NBS), [40].

The contribution rates of the intraregional Theil index to the total Theil index between urban and rural areas were calculated and are shown in Tables 4 and 5.

**Table 4.** The four regions' intraregional contribution rate in rural areas, 2010–2019.

| Rural | East | Northeast | Central | West |
|---|---|---|---|---|
| 2010 | 18.989% | 3.147% | 13.140% | 18.239% |
| 2011 | 18.433% | 2.868% | 13.712% | 19.371% |
| 2012 | 18.148% | 2.928% | 14.535% | 19.422% |
| 2013 | 18.016% | 1.687% | 13.325% | 17.662% |
| 2014 | 19.491% | 1.542% | 18.366% | 17.688% |
| 2015 | 24.384% | 1.647% | 18.935% | 18.589% |
| 2016 | 24.420% | 1.696% | 18.466% | 19.028% |
| 2017 | 24.372% | 1.763% | 17.703% | 18.572% |
| 2018 | 25.092% | 1.867% | 17.198% | 18.588% |
| 2019 | 25.547% | 1.718% | 16.941% | 18.815% |

Source: Authors own calculation using data source from China's National Bureau of Statistics, (NBS), [40].

**Table 5.** The four regions' intraregional contribution rate in urban areas, 2010–2019.

| Urban | East | Northeast | Central | West |
|---|---|---|---|---|
| 2010 | 33.036% | 3.638% | 5.909% | 8.518% |
| 2011 | 32.517% | 3.514% | 7.049% | 9.227% |
| 2012 | 32.963% | 3.291% | 7.194% | 10.019% |
| 2013 | 32.395% | 3.084% | 7.988% | 8.702% |
| 2014 | 41.175% | 4.257% | 7.769% | 7.428% |
| 2015 | 46.218% | 4.139% | 8.261% | 6.369% |
| 2016 | 45.691% | 4.070% | 8.351% | 6.092% |
| 2017 | 44.986% | 3.876% | 8.439% | 6.172% |
| 2018 | 45.125% | 3.845% | 8.569% | 6.105% |
| 2019 | 33.025% | 20.477% | 6.429% | 14.836% |

Source: Authors own calculation using data source from China's National Bureau of Statistics, (NBS), [40].

The results show that in each region, the size and variation trends of the intraregional Theil index are different between rural and urban areas. For rural areas, the Theil index in Figure 8 showed a trend of declining first and then rising in the eastern region, declining continuously from 2010 to 2015, and rising from 2016 to 2019, indicating that the income gap between prefecture-level cities in the eastern region has been expanding. In the central region, the Theil index showed a trend of rising first and then falling, with 2015 as the inflection point, but the range of change is relatively small, which means that the income gap in the central region is relatively stable. The Theil index in the northeast and west declined overall from 2010 to 2019, but it fell more sharply and at a faster rate in the northeast during 2012–2014, by about a half. Among the four regions, the western region had the highest Theil index in 2010, but the eastern region replaced it as the highest in 2015. The Theil index in the northeast is the smallest and the central region has the Theil index between the west and northeast. In terms of contribution rate to the total Theil index, shown in Table 4, the contribution rates of eastern and western regions are the largest, followed by the central region and then the northeastern region with the smallest.

For urban areas, the Theil index of the eastern region declined from 2010 to 2013, and then significantly rose from 0.020 in 2013 to 0.042 in 2019, as shown in Figure 9. This means that the income inequality between cities in the eastern region has increased significantly during the recent period. In the northeast, the trend was similar but smaller, and the tipping point was delayed to 2014, indicating that the intercity inequality has not grown as quickly as in the east. The Theil index in the central region fluctuated slightly but remained relatively stable at around 0.011. Income inequality in the western region has fluctuated considerably. It showed a significant decline from 0.017 in 2010 to 0.011 in 2018 but rose significantly to 0.017 in 2019. Among the four regions, inequality is most significant in the

east, followed by the northeast. While the western region was once below the central region in the Theil index, the central region showed the smallest and most stable intraregional variation in 2019. Considering the contribution rate, shown in Table 5, the contribution to the Theil index is the largest in the eastern region, followed by the northeast and the west, and the least in the central region.

The study of income inequality shifted from an international to intranational scale with the improvement of spatial analysis [34,46]. There are several methods to analyse spatial inequality within and between groups in one country at a certain period, such as the Gini coefficient decomposition [47,48], the Sharpley value decomposition [49,50], and the Theil decomposition [20].

In this work, we chose the one-stage Theil decomposition to decompose total income inequality into within-region inequality and between-region inequality. Compared with the other two methods, there are relatively few empirical studies that use Theil decomposition to analyse regional income disparities in China. Since the Theil decomposition method could be used to analyse discontinuous cross-sectional data, most research using Theil decomposition uses microscopic household data with several waves [51,52]. For studies that use macro data to measure income inequality, Tian et al. (2016) grouped China's provinces into two groups according to the time of economic convergence (primary and latter) and found that within-group income inequality has declined while between-group income inequality has increased over time [15]. This grouping definition would change over time; it is more common to divide provinces into groups based on geo-economic location [53,54]. In this paper, we used continuous macro-economic data from NBS and four geo-economic regions to investigate the within- and between-region disparity from 2010 to 2019.

Akita (2003) used the two-stage Theil decomposition method to calculate the within-province inequality in China by taking the income levels of the lower boroughs of a province. This reflects deeper regional disparities in China [20]. However, due to the limitation of administrative division, municipalities with no lower-level cities are not included in the research scope. Therefore, we used per capita income data from 271 cities, including municipalities, and one-stage Theil decomposition for a more comprehensive measure of regional inequality among urban and rural residents.

To further investigate an explanation for the trend in the change in income inequality in China, we examined the factors that influence income inequality. According to Kuznets Theory, the urbanization rate, GDP growth rate, education growth rate, higher education rate, fiscal spending/GDP, and FDI/GDP were chosen as the influencing factors to reflect the macro-environmental changes.

### 3.4. Factors Influencing Income Inequality

To better measure the factors affecting income inequality, we first ran a simple linear regression using the mixed OLS model. The fixed effects and random effects models were found to outperform the mixed OLS model after testing for individual and time effects. According to the results of Hausmann regression, we concluded that the optimal model was the fixed effects model (Table 5).

The comparison results of the three models are shown in Table 6. It can be seen from the regression results that urbanization rate, GDP growth rate, fiscal spending rate, and FDI rate are significant in all three regression models, which means that these indicators are robust in the regression results. In the selected fixed effects model, the impact of urbanization rate, GDP growth rate, and fiscal spending rates are relatively significant, having strong explanatory power for income inequality.

The regression coefficients of urbanization rate and fiscal spending rates are significantly negative within a 99% confidence interval, indicating that a higher level of urbanization and more government fiscal spending are beneficial to reducing the income disparities between regions. The regression coefficient of GDP growth rate is also significantly positive within a 99% confidence interval, which means local economic development may come at

the expense of income equality. Furthermore, the influence of FDI on regional disparity is negative within a 95% confidence interval. Within the transition of FDI from coastal to inland regions, the increase in FDI from 2000 to 2020 is positively associated with the regional income convergence. In this empirical model, the effect of education year growth and higher education proportion on the inequality is less significant than other factors.

**Table 6.** The three models' estimation results.

|  | **Mixed OLS** | **Fixed Effects** | **Random Effects** |
|---|---|---|---|
| Urbanization rate | −0.306 *** (0.018) | −0.216 *** (0.018) | −0.262 *** (0.018) |
| GDP | 0.182 *** (0.021) | 0.119 *** (0.015) | 0.118 *** (0.015) |
| EDU | 0.021 (0.031) | 0.007 (0.021) | 0.002 (0.022) |
| HEDU | 0.075 ** (0.035) | 0.056 * (0.033) | 0.040 (0.034) |
| FISCAL | 0.020 *** (0.008) | −0.153 *** (0.014) | −0.096 *** (0.012) |
| FDI | −0.320 *** (0.072) | −0.162 ** (0.065) | −0.236*** (0.066) |
| _cons | 0.248 *** (0.007) | 0.250 *** (0.008) | 0.263 *** (0.009) |
| N | 651.000 | 651.000 | 651.000 |
| r2 | 0.726 | 0.636 | |
| r2_a | 0.723 | 0.615 | |
| | | | Prob > chi2 = 0.0000 |

Standard errors in parentheses. * $p < 0.1$, ** $p < 0.05$, *** $p < 0.01$. Source: Authors own calculation using data source from China's National Bureau of Statistics, (NBS), [40].

In this paper, we focus on estimating the impact of different factors on four regions and put forward corresponding policy implications. To investigate the impact of each driver on different areas, we further carried out grouping regression of fixed effects for provinces in four regions, and the regression results are shown in Table 7. By grouping regression, we can obtain regression results for the four regions.

*3.5. Robustness of Data and Model*

Conducting robustness tests is crucial in ensuring the reliability and validity of research findings. By subjecting our study to various tests, we can identify potential issues with data or model specifications and make necessary adjustments to improve the credibility of our results.

In our study, we performed tests for multicollinearity, normality, and heteroscedasticity to address concerns that may affect the accuracy of our results. By addressing these issues, we can be confident that our findings are not spurious or biased due to data or model misspecification.

Furthermore, by using different econometric models, such as the mixed OLS regression model, fixed effects model, and random effects model, we can further test the robustness of our results under different assumptions and specifications. This helps to ensure that our conclusions are not overly reliant on the choice of a particular model and can be generalized to a wider range of settings.

In essence, the purpose of conducting robustness tests is to provide a rigorous and trustworthy analysis that can withstand scrutiny and contribute to the advancement of knowledge in the field. By doing so, we can increase confidence in the results of our study and contribute to the overall understanding of the phenomenon being studied.

**Table 7.** Grouped regression results.

|  | Total | Central | East | Northeast | West |
|---|---|---|---|---|---|
| URBAN | −0.216 *** | −0.211 *** | −0.113 *** | 0.143 | −0.307 *** |
|  | (0.018) | (0.061) | (0.020) | (0.107) | (0.050) |
| GDP | 0.119 *** | 0.143 *** | 0.119 *** | 0.042 * | 0.125 *** |
|  | (0.015) | (0.021) | (0.017) | (0.022) | (0.031) |
| EDU | 0.007 | 0.003 | 0.013 | 0.083 | 0.002 |
|  | (0.021) | (0.039) | (0.030) | (0.058) | (0.032) |
| HEDU | 0.056 * | −0.287 *** | 0.051 ** | −0.124 | 0.008 |
|  | (0.033) | (0.108) | (0.025) | (0.114) | (0.107) |
| FISCAL | −0.153 *** | 0.272 *** | −0.057* | −0.165 *** | −0.150 *** |
|  | (0.014) | (0.084) | (0.031) | (0.042) | (0.019) |
| FDI | −0.162 ** | 0.154 | 0.099* | 0.239 ** | −0.605 ** |
|  | (0.065) | (0.224) | (0.054) | (0.112) | (0.263) |
| _cons | 0.250 *** | 0.165 *** | 0.127 *** | 0.039 | 0.339 *** |
|  | (0.008) | (0.013) | (0.012) | (0.053) | (0.015) |
| N | 651.000 | 126.000 | 210.000 | 63.000 | 252.000 |
| r2 | 0.636 | 0.721 | 0.580 | 0.694 | 0.720 |
| r2_a | 0.615 | 0.694 | 0.548 | 0.649 | 0.700 |

Standard errors in parentheses. * $p < 0.1$, ** $p < 0.05$, *** $p < 0.01$. Source: Authors own calculation using data source from China's National Bureau of Statistics, (NBS), [40].

## 4. Discussion

Income inequality is an important issue for every country, and it remains at a relatively high level globally [55]. In recent years, this issue has received increasing attention, particularly as developing countries have experienced economic development [7,56,57]. Some studies suggest that income inequality in the developing world has decreased over the past 30 years, but this is primarily due to a reduction in inequality between countries [56]. Within developing countries, average inequality has actually been slowly rising [56,58]. However, countries with leftist government parties have some small income gaps from the 1970s to the beginning of the 21st century [58]. Moreover, higher growth rates in average incomes have not led to a reduction in inequality within countries [55,59]. Although economic growth can help reduce absolute poverty, this effect is often less significant in countries with high levels of inequality [56].

Previous studies in China show that urban bias in policy contributes to the widening regional income disparity, through which regions with a higher urbanization rate benefit more than those with a lower urbanization rate [35,60]. The effect of urbanization on income convergence is the largest in the western region and the weakest in the eastern region. Compared with eastern and central provinces, the urbanization level in the west is relatively low [61]. As the Chinese government accelerates urbanization, the growth rate in the west is obviously faster than that of other regions with a larger base. However, the rural-to-urban migration in northeast China is not obvious because of the serious population loss. This improvement in urbanization is beneficial to increase resident income and reduce the income gap [62,63]. According to our empirical results, the urbanization rate has a significant negative impact on inequality in the eastern, central, and western regions, but the impact is not significant in the northeast region from 2010 to 2019. Therefore, there are positive effects on income convergence of promoting the urbanization of the population—although income inequality between urban and rural areas may be increasing, the share of overall income inequality is decreasing.

According to our research, economic growth has intensified income disparities in all four regions. Economic growth in the northeast has had the least impact on income inequality compared with other regions. This indicates that China's economic development over the past two decades, to some extent, has taken income equity as payment [12,16]. Research

also shows that income inequality has had a negative impact on economic development in China since the 1990s [55,60]. X. Luo and Zhu (2008) suggest that rising disparity is part of the normal process of economic development at a certain stage [56]. By analysing data from developed and developing countries, Shin (2012) found that higher income disparity tends to retard growth in the early stage of economic development, until achieving growth in a near-steady state [57].

The impact of the proportion of the population with higher education on income inequality is different in central and eastern regions. In the east, where income levels are higher, the increase in the proportion of people with higher education has a positive impact on income inequality. This is similar to several developed countries, where a more abundant labour market and trade increase differentials in returns to education and skill [18,31,38]. On the contrary, empirical results show that this higher education proportion has contributed to the income convergence in central China between 2000 and 2020. One possible explanation is that individuals with a higher education level have more advantages in gaining human capital revenue. As the proportion of the well-educated population increases, income converges under the influence of human capital improvement [30].

After the rapid growth of income inequality in China in the 1990s, the central government adopted a series of policies in the 21st century to alleviate regional inequality. To narrow the gap between the east and other regions, the Chinese government launched a series of programs in the west, northeast, and central China. These include the "great Weser Development", "Rejuvenating Northeast Old Industrial Bases", and "the Rise of Central China". The investment of these programs could be reflected in fiscal spending as a share of GDP. According to the results of empirical research, fiscal expenditure plays a key role in reducing income disparity in the west and northeast (both regression coefficient indices are negative and significant). Furthermore, Sicular et al. (2008) believe that policies implemented in the country's regions to narrow the urban–rural gap, such as the "Build a Socialist New Countryside" campaign of 2006, played a positive role in reducing the western income inequality [58]. The positive impact of fiscal spending on income disparity in the central region may be due to the governments in this region making more efforts to develop industries that require advanced degrees. There are few related studies on this, and the results need to be further investigated.

There are studies that use FDI as an indicator to measure the impact of globalization on income inequality [59,64]. Wan et al. (2007) argued that the impact of globalization on China's regional income inequality was growing in the early stage of reform and opening up [18]. According to our empirical results, FDI indeed has a positive impact on disparity, but it has been weakened as China's openness has reached a relatively high level; even in the central region, the impact has been insignificant.

## 5. Conclusions

In this research, we analysed the current situation and characteristics of income inequality in the eastern, northeastern, central, and western regions of China. Based on the urban and rural income data of 271 prefecture-level cities, the total Theil index was calculated with population as the weight, as well as the regional differences in income inequality in four regions from 2010 to 2019.

From the regional characteristics of income inequality, there are significant differences in income inequality among 31 provincial administrative regions in China. During the observation period (2000–2020), the general trend was for income inequality to first increase and then decrease. Specifically, the western region has the highest income inequality index, and the rate of decline is relatively fast. The differentiation of income level in eastern China is obvious. According to the results of the Theil index, the income inequality of different regions in China has a declining trend in rural areas, and a declining and then increasing trend in urban areas. From the perspective of influencing factors, local economic

development has a positive impact on income inequality, while the urbanization rate and fiscal spending rate have negative impacts on income inequality.

First, the increased investment in human capital, which mainly refers to investment in education, has reduced income inequality. The increase in the level of education in China shows a more concentrated situation in the eastern and urban areas. The rural areas and the central, western, and northeastern regions have not been able to benefit from the same educational resources as the east and urban areas. According to our research, education is an important factor in raising income levels. The lower level of education and the poorer quality of education affect the labour efficiency of the inhabitants of these regions. Therefore, there should be more investment in education in the central and western regions to increase the parity of educational resources nationwide.

Secondly, economic growth in general has widened income inequalities. Accelerating the relative growth rate of the central and western regions will widen income inequality within the regions, so how can economic growth be accelerated? The government should always be concerned with balancing efficiency and equity in the process of economic development and finding the comparative advantages of the region. For example, the central and western regions often have lower labour costs. Foreign investment, as well as investment from the east, could boost these regions' development. At the same time, sound infrastructure such as transport and communications can reduce transaction costs and thus speed up economic development.

### 6. Study Limitations

This study has several limitations that should be taken into consideration when interpreting its findings. First, the study relied on publicly available data from the National Bureau of Statistics of China, which may have limitations in terms of accuracy and comprehensiveness. Although efforts were made to ensure the quality of the data, the reliability of the results may be affected by potential errors or biases in the data.

Secondly, the study used the Theil index as the primary measure of income inequality. While the Theil index is a widely used and recognized measure, it has certain limitations. For example, it may not capture the full extent of income inequality in regions with significant outliers, and it may be affected by changes in the overall level of income. Therefore, caution should be exercised when interpreting the results, and alternative measures of income inequality should be considered in future research.

In summary, while this study provides valuable insights into the patterns and determinants of income inequality in China, its findings should be interpreted with caution due to the limitations of the data and the measures used. Further research is needed to address these limitations and to provide a more comprehensive understanding of income inequality in China.

**Author Contributions:** Conceptualization, X.Y.; Data curation, X.Y. and S.M.; Formal analysis, X.Y.; Funding acquisition, S.M.; Methodology, X.Y. and S.M.; Project administration, S.M.; Software, X.Y.; Supervision, S.M.; Visualization, S.M.; Writing—original draft, X.Y.; Writing—review and editing, S.M. All authors have read and agreed to the published version of the manuscript.

**Funding:** No funding was received for this paper.

**Institutional Review Board Statement:** Not applicable as this study used secondary data.

**Informed Consent Statement:** Not applicable as this study used secondary data.

**Data Availability Statement:** Not applicable.

**Conflicts of Interest:** The authors declare no conflict of interest.

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
