# Peer review of "Trends and Causes of Regional Income Inequality in China"

_sustainability, doi:10.3390/su15097673_

Round 1

Reviewer 1 Report

Thanks for giving this interesting paper. I think the authors do a very good job. There are some points concerning this article as well for your consideration. 

1. In the subsection of "2.3 Data"

I am looking forward to the definition and statistic descriptions of all variables. I think it would be more clear to give an extensive definition of each abbreviation at the beginning. 

2.  About the model

I am so curious about two variables in the model, that is, "Edu growth rate" and "Higher edu rate". Is there any possible collinearity problem between them?  

3. in the sub-section of "3.2 Urban and rural disparity" 

I wonder if it will present some interesting regression results to address the determinants of income inequality among urban and rural areas across different regions. However, it is not. I am still interested in that kind of result, which can be more interesting for the paper's contribution. 

Author Response

Comment

Response

Change in the paper

Thanks for giving this interesting paper. I think the authors do a very good job. There are some points concerning this article as well for your consideration. 

Thank you for this encouraging comment.

1. In the subsection of "2.3 Data"

I am looking forward to the definition and statistic descriptions of all variables. I think it would be more clear to give an extensive definition of each abbreviation at the beginning. 

Thank you for this comment. We have included a table that detail out the data and descriptions.

Table 2.

2.  About the model

I am so curious about two variables in the model, that is, "Edu growth rate" and "Higher edu rate". Is there any possible collinearity problem between them?  

Provide definition of EDU and HIGHER EDU - take the formal definition.

Provide the test of collinearity for the two variables.

The explanation of two variables have been added into Table 2. Besides,  the VIF test showed that there is no co-linearity for the two variables.

3. in the sub-section of "3.2 Urban and rural disparity" 

I wonder if it will present some interesting regression results to address the determinants of income inequality among urban and rural areas across different regions. However, it is not. I am still interested in that kind of result, which can be more interesting for the paper's contribution. 

The effect of urban and rural is included in the regression estimation as one of the independent variables. URBAN represents urban = 1 and rural = 0. At present, we feel that this is adequate, we value the suggestion by the reviewer and would consider this for our next publication.

Reviewer 2 Report

Referee Report

“Trends and Causes of Regional Inequality in China”

This paper does two things. First it is a calculation of regional income inequality in China using the Theil Index. Second, it uses panel data analysis to explain inequality from 2010 to 2019.  Inequality and its determinants are a growing literature in a number of academic disciplines.  My comments below should be taken in the spirit of improvement.

Major comment #1:  Number of Provinces and Number of Observations

In Table 1 the authors state that they break down the provinces into four regions. They state in the text (page 5, line 168) that there are 31 provinces but in Table 1 there are only 30 listed. This is important not only because it is contradictory, but also because I need help clarifying the number of observations in the panel regressions.  In Table 4 the authors state that N=651.  Yet the authors state that they are covering the period from 2010 to 2019. This would mean the number of years is equal to 10 and the number of cross-sectional units is either 31 or 30. That would leave the authors with 300 or 310 observations if they are running a balanced panel. 

·       Please explain the total number of provinces included.

·       Enumerate the total number of observations in the regressions and how it came to be 561. If this in error, what is the correct number?

·       If it is not a balanced panel, please explain why and what data are missing.

Major comment #2: Determinants of Inequality – Theory and Measurement

The authors just present regressions regarding the determinants of income inequality but without justifying those variables based on theory and the literature. They also do not define the data. For example, what is edu growth, how is it different than Higher Edu rate and how highly correlated are these two variables?

In addition, is there that much year-to-year variation in your independent variables? Inequality – like many macro level variables – evolves slowly. Typically in cross-country growth regressions, for example, you see five-year panels for this reason.  Otherwise you are just artificially inflating your standard errors with data that provide more degrees of freedom but not a lot of information.  If education is a stock variable, for example, then it will not change very much year-to-year.

Note that you may be trying to say you are inspired by two papers in this sentence, but it is not clear:

As for the influence factors of disparity, existing studies mostly focus on the impact 402 of one factor or several factors on national income inequality [31,47].”

You should define each variable separately and explain the expected relationship based on theory and the literature.

Major comment #3: Robustness Checks

The authors treat their results as causal (see, for example, their discussion section), but they have no identification strategy. They treat their panel regressions and specifications as “true” even though there is not a structural model to say why that is the model they should estimating.  Before making policy claims, the authors should demonstrate how their results are stable their results are to the exclusion of some variables.  In particular, given the importance they place on their results when they gain or lose significance, they need to show that these results are robust.

Major comment #4: Why Split into Regions?

The authors never defend the splitting of the sample in Table 5 into the 4 regions. A Chow test should be performed to show that these 4 areas are actually different enough to be estimated separately.  And note that one of your regions only has three provinces so that specification is really limited.

Major comment #5 – Why sustainability?

This paper was submitted to Sustainability. What do the results have to do with sustainability? The authors do not highlight how inequality is not sustainable. 

Major comment #6 – Editorializing

The authors editorialize a lot in a manner not backed up by literature or their results. For example, they state that inequality is a feature of a market economy.  It is actually a feature of all economies. See David Henderson’s work on inequality in the Soviet Union based on archival records.  Those in the party had much higher standards of living than the average citizen.  Similarly, the same is true with Cuba.

The same is true of much of the policy advice in the discussion.  Where does a discussion of capital movement come from and how is it supported by the results? And why not the free movement of people? That should lead to income convergence across areas?

Major comment #7 – Writing

A lot of the sentences are unclear and could benefit from a copyeditor.

For example, line 416: “The effect of urbanization on income convergence is the largest in the western groin and the weakest in the eastern region.”

Line 315: “The results show that the size and variation trend of intra-region the Theil index are different in each region between rural and urban area.”

Author Response

Comment

Response

Change in the paper

This paper does two things. First it is a calculation of regional income inequality in China using the Theil Index. Second, it uses panel data analysis to explain inequality from 2010 to 2019.  Inequality and its determinants are a growing literature in a number of academic disciplines.  My comments below should be taken in the spirit of improvement.

Major comment #1:  Number of Provinces and Number of Observations

In Table 1 the authors state that they break down the provinces into four regions. They state in the text (page 5, line 168) that there are 31 provinces but in Table 1 there are only 30 listed.

Our apology for this confusion. Yes, there is 31 provinces and we have left out Tibet This has been added to the paper.

Table 1

This is important not only because it is contradictory, but also because I need help clarifying the number of observations in the panel regressions.  In Table 4 the authors state that N=651.  Yet the authors state that they are covering the period from 2010 to 2019. This would mean the number of years is equal to 10 and the number of cross-sectional units is either 31 or 30. That would leave the authors with 300 or 310 observations if they are running a balanced panel.  

Our apology for this confusion. There are 651 observations. The regression analysis uses data from the year 2000 - 2020.

The years 2010-2019 were used for theil index calculation by cities.

The information on this has been amended for clarity.

2.3 Data

Please explain the total number of provinces included.

There are 31 provinces in the data.

Enumerate the total number of observations in the regressions and how it came to be 561. If this in error, what is the correct number?

There are 31 provinces with 21 years, hence 651 observations for the regression analysis.

If it is not a balanced panel, please explain why and what data are missing.

The estimate is a balanced panel estimation.

Major comment #2: Determinants of Inequality – Theory and Measurement

The authors just present regressions regarding the determinants of income inequality but without justifying those variables based on theory and the literature. They also do not define the data. For example, what is edu growth, how is it different than Higher Edu rate and how highly correlated are these two variables?

The data has been defined in table 2.

The theory that was adopted in the study is Kuznet’s curve as shown in section 2.2.

It has been mentioned that in the paper that 

Based on the Kuznets Theory, the income inequality in one country is associated with economic growth, human capital development and financial condition improvement

The variables used in the estimation correspond to the theory that are GDP and URBAN (economic growth) EDU and HEDU (human capital development) and FISCAL and FDI (financial condition)

In addition, is there that much year-to-year variation in your independent variables? Inequality – like many macro level variables – evolves slowly. Typically in cross-country growth regressions, for example, you see five-year panels for this reason.  Otherwise you are just artificially inflating your standard errors with data that provide more degrees of freedom but not a lot of information.  If education is a stock variable, for example, then it will not change very much year-to-year.

Yes, there is variability in the data. A table on the data descriptive statistics is provided in Table 3.

Table 3.

Note that you may be trying to say you are inspired by two papers in this sentence, but it is not clear:

As for the influence factors of disparity, existing studies mostly focus on the impact 402 of one factor or several factors on national income inequality [31,47].”

This sentence has been deleted to avoid confusion.

You should define each variable separately and explain the expected relationship based on theory and the literature. 

This has been added in Table 2.

Table 2

Major comment #3: Robustness Checks

The authors treat their results as causal (see, for example, their discussion section), but they have no identification strategy. They treat their panel regressions and specifications as “true” even though there is not a structural model to say why that is the model they should estimating.  Before making policy claims, the authors should demonstrate how their results are stable their results are to the exclusion of some variables.  In particular, given the importance they place on their results when they gain or lose significance, they need to show that these results are robust.

The estimation follows the theory of Kuznets Curve as mentioned in the paper.

Based on the Kuznets Theory, the income inequality in one country is associated with economic growth, human capital development and financial condition improvement

The variables used in the estimation correspond to the theory that are GDP and URBAN (economic growth) EDU and HEDU (human capital development) and FISCAL and FDI (financial condition)

Robustness of the data has been included in the paper

Robustness of data and model

Major comment #4: Why Split into Regions?

The authors never defend the splitting of the sample in Table 5 into the 4 regions. A Chow test should be performed to show that these 4 areas are actually different enough to be estimated separately.  And note that one of your regions only has three provinces so that specification is really limited.

We provided an explanation on the different characteristics of the different regions that merit investigation of the four regions. A sentence has been added in line 106 to further strengthen the analysis into regions.

Line 95-106

Major comment #5 – Why sustainability? 

This paper was submitted to Sustainability. What do the results have to do with sustainability? The authors do not highlight how inequality is not sustainable.  

Explanation has been added in the introduction that link income inequality to sustainability.

Major comment #6 – Editorializing

The authors editorialize a lot in a manner not backed up by literature or their results. For example, they state that inequality is a feature of a market economy.  It is actually a feature of all economies. See David Henderson’s work on inequality in the Soviet Union based on archival records.  Those in the party had much higher standards of living than the average citizen.  Similarly, the same is true with Cuba.

This statement was gathered from journal article by Luo, Li, & Sicular (2020).

However, since this is confusing, we have changed this statement to the work of David Henderson.

According to Henderson [3], inequality persists in all economies, even communist ones like the Soviet Union. He also draws attention to how political influence and resource distribution play a part in creating patterns of inequality within a society.

Line 33-36

The same is true of much of the policy advice in the discussion.  Where does a discussion of capital movement come from and how is it supported by the results? And why not the free movement of people? That should lead to income convergence across areas?

This represents the current situation in China.

Nevertheless, as the write up has little relation with the paper findings, this sentence has been deleted.

Major comment #7 – Writing

A lot of the sentences are unclear and could benefit from a copyeditor.

For example, line 416: “The effect of urbanization on income convergence is the largest in the western groin and the weakest in the eastern region.”

Line 315: “The results show that the size and variation trend of intra-region the Theil index are different in each region between rural and urban area.”

The paper will be sent for English editing by MDPI.

Reviewer 3 Report

The title of the article does not match the content of the article. The essence of the article is income inequality, so this should also appear in the title of the article. Even though it is an article about China, there is not enough literature about other countries in the literature. Comparison of income inequality with other European or Asian countries.

Some pictures are very small and illegible, and some pictures are not referenced in the text.

The discussion could be linked directly to the results, and there is a lack of more specific comparisons with other researches in this area.

In the end, the limits of the research should be revealed.

Author Response

Comment

Response

Change in the paper

The title of the article does not match the content of the article. The essence of the article is income inequality, so this should also appear in the title of the article. Even though it is an article about China, there is not enough literature about other countries in the literature. Comparison of income inequality with other European or Asian countries.

The word ‘income’ has been added to the title

Trends and Causes of Regional Income Inequality in China

Some pictures are very small and illegible, and some pictures are not referenced in the text.

Thank you for your comment. We take note of the comment. The actual size picture has been shared as supplementary files. Nevertheless, we have changed all the pictures in the document.

All figures.

The discussion could be linked directly to the results, and there is a lack of more specific comparisons with other researches in this area.

More literature of other countries is included in the first paragraph in discussion.

Discussion section

In the end, the limits of the research should be revealed.

Limitation has been added.

Limitation of this paper has been added.

Round 2

Reviewer 2 Report

I appreciate the verbal statement on lines 94-106 that the regions are different. However, I requested a statistical test as to the differences between regions. A Chow test should tell you if the regions should be segmented for the purposes of your regressions.  For an example, see:

Brasington, David M. "Differences in the production of education across regions and urban and rural areas." Regional Studies 36, no. 2 (2002): 137-145.

Author Response

We have amended the paper according to the reviewer's comments as described in the response page. 

Reviewer 3 Report

Tables 2 - 7, and figures 5 - 9 do not have a source listed

Author Response

We have amended the paper following the reviewer's comments as shown in the response file. 
